# Vitamin K2 Protects Against SARS-CoV-2 Envelope Protein-Induced Cytotoxicity in Chronic Myeloid Leukemia Cells and Enhances Imatinib Activity

**DOI:** 10.3390/ijms252111800

**Published:** 2024-11-02

**Authors:** Seiichi Okabe, Yuya Arai, Akihiko Gotoh

**Affiliations:** Department of Hematology, Tokyo Medical University, Tokyo 160-0023, Japan; y-arai@tokyo-med.ac.jp (Y.A.); akgotou@juntendo.ac.jp (A.G.)

**Keywords:** SARS-CoV-2, envelope protein, vitamin K2, CML, imatinib

## Abstract

Chronic myeloid leukemia (CML) is a myeloproliferative neoplasm characterized by excessive proliferation of myeloid cells. The COVID-19 pandemic has raised concerns regarding the impact of SARS-CoV-2 on patients with malignancies, particularly those with CML. This study aimed to investigate the effects of SARS-CoV-2 proteins on CML cell viability and the protective role of vitamin K2 (VK2) in conjunction with imatinib. Experiments conducted on K562 CML cells demonstrated that the SARS-CoV-2 envelope protein induces cytotoxicity and activates caspase 3/7, which are key markers of apoptosis. VK2 mitigated these cytotoxic effects and decreased cytokine production while inhibiting colony formation. Furthermore, the combination of VK2 with imatinib significantly reduced cellular proliferation, diminished mitochondrial membrane potential, and markedly suppressed colony formation. These findings suggest that VK2 protects CML cells from SARS-CoV-2-induced cytotoxicity and enhances the therapeutic efficacy of imatinib, presenting a potential strategy to improve CML treatment during the COVID-19 pandemic.

## 1. Introduction

SARS-CoV-2, the causative agent of COVID-19, is a highly transmissible and pathogenic coronavirus first identified in Wuhan, China, in December 2019. This novel virus, which causes severe acute respiratory syndrome, rapidly escalated into a global pandemic, constituting a major public health emergency [1]. Early studies demonstrated that the receptor-binding domain (RBD) of SARS-CoV-2 exhibits significantly greater affinity for the human angiotensin-converting enzyme 2 (ACE2) receptor compared to SARS-CoV, facilitating enhanced viral entry and replication [2]. The SARS-CoV-2 genome encodes four structural proteins—Spike (S), Envelope (E), Membrane (M), and Nucleocapsid (N), along with 14 open reading frames (ORFs) that encode 27 non-structural and accessory proteins. SARS-CoV-2 shares approximately 80% genomic sequence homology with other human coronaviruses [1]. Clinically, COVID-19 often manifests as viral pneumonia and can progress to acute respiratory distress syndrome (ARDS) in severe cases. COVID-19 has caused a significant increase in hospitalizations for pneumonia with multiorgan disease, compromising public health and medical services around the world [3]. Biomarkers, including serum albumin, lactate dehydrogenase (LDH), C-reactive protein (CRP), lymphocyte percentage, and neutrophil-to-lymphocyte ratio (NLR) have been identified as prognostic indicators of disease severity [4]. Furthermore, the introduction of vaccines against SARS-CoV-2 has changed the course of the pandemic [5]. In fact, the recent development of vaccines is considered a powerful measure that saved lives and minimized the impact on health, social systems, and global economics [6].

During the COVID-19 pandemic, hematological abnormalities, such as leukocytosis, leukopenia, and lymphopenia, were frequently observed [7]. Research indicates that SARS-CoV-2 exerts profound effects on both the hematopoietic and hemostatic systems; however, the underlying mechanisms of COVID-19-induced leukopenia remain poorly understood [8]. The proposed pathophysiological mechanisms include ACE2-mediated lymphopenia due to direct viral invasion of lymphocytes, apoptosis induced by pro-inflammatory cytokines, and possible viral infiltration of lymphoid tissues [9].

Chronic myeloid leukemia (CML) is a clonal myeloproliferative neoplasm characterized by t(9;22)(q34;q11.2) reciprocal translocation, resulting in the fusion of the Abelson gene (*ABL1*) on chromosome 9q34 with the breakpoint cluster region (*BCR*) gene on chromosome 22q11.2, forming the *BCR::ABL1* fusion gene [10]. This translocation, commonly referred to as the Philadelphia chromosome, encodes the constitutively active BCR-ABL tyrosine kinase, a major driver of leukemogenesis [11]. The introduction of BCR-ABL1 tyrosine kinase inhibitors (TKIs) has markedly improved the prognosis of CML [12]. Imatinib, a first-generation ABL1 TKI, was approved as first-line therapy following the landmark International Randomized Study of Interferon and STI571 (IRIS trial) for newly diagnosed patients with CML [13]. Despite the efficacy of imatinib and second-generation TKIs, such as nilotinib and dasatinib, resistance to TKI therapy remains a challenge in a subset of patients, necessitating alternative therapeutic strategies [14].

Vitamin K (VK), a fat-soluble vitamin, is essential for several physiological functions and exists in two primary forms: phylloquinone (K1) and menaquinone (K2) [15]. VK is integral to the regulation of coagulation, inhibition of vascular calcification, maintenance of bone homeostasis, and modulation of cellular proliferation and apoptosis [16]. However, menaquinones (MK) and phylloquinones differ in their biological activity, exhibiting variations in bioavailability, plasma half-life, and transport mechanisms. VK1 and MK-4 are detectable in plasma for 8–24 h, whereas long-chain menaquinones can persist for up to 96 h post-administration [16]. Both VK1 and VK2 are crucial for the post-translational γ-carboxylation of glutamic acid residues in VK-dependent proteins, including coagulation factors. These modified proteins are implicated in diverse physiological and pathological processes, such as oncogenesis [17]. Previous studies have highlighted the potential therapeutic effects of VK in the management of infections, inflammatory diseases, and autoimmune disorders [18].

While significant progress has been made in reducing COVID-19-related mortality through widespread vaccination and the availability of effective antiviral therapies [4], the specific risk profile of patients with CML infected with SARS-CoV-2 remains uncertain. This study aimed to investigate the effects of SARS-CoV-2 on CML cells in vitro, with a particular focus on the potentially protective role of VK2 in mitigating SARS-CoV-2-induced cytotoxicity and enhancing the efficacy of ABL1-TKIs. Our findings indicate that VK2 attenuates SARS-CoV-2-induced cytotoxicity in CML cells and augments the therapeutic efficacy of imatinib.

## 2. Results

### 2.1. Gene Expression Analysis of COVID-19-Related Genes

An initial gene expression analysis was conducted by comparing datasets from patients with CML to those from healthy controls that were retrieved from the Gene Expression Omnibus (GEO) database. A comprehensive volcano plot illustrating the differentially expressed genes is shown in Figure 1A. Additionally, a dendrogram representing hierarchical clustering was generated from the same database, revealing distinct gene expression profiles (Figure 1B). Gene Ontology (GO) analysis, a system used to categorize gene functions and cellular locations [19], identified upregulation of immune response-related processes. Pathway enrichment analysis highlighted the key signaling networks implicated in SARS-CoV-2 infection (Figure 1C). Analysis of the GEO database revealed elevated expression levels of pro-inflammatory cytokine genes, including interleukin-1α *(IL-1A*), *IL-1β*, and *IL-6*, in individuals with severe COVID-19 compared to healthy controls. However, the expression of tumor necrosis factor (*TNF*) messenger RNA (mRNA) remained unchanged in patients with COVID-19 compared to controls (Figure 1D).

### 2.2. Effect of the SARS-CoV-2 Envelope Protein on CML Cells

SARS-CoV-2 encodes four structural proteins [1]. This study investigated the potential mechanisms by which the S and E proteins contribute to COVID-19 pathogenesis in CML cells. The functions of the S1 subunit of the Spike and E proteins were examined in the CML K562 cell line. Treatment with the E protein (EP) induced a dose-dependent reduction in K562 cell viability, whereas exposure to the S1 protein showed no significant cytotoxic effects (Figure 2A). EP-induced cytotoxicity was prominent, whereas S1 exhibited no inhibitory effects on K562 cells (Figure 2B), indicating differential cellular responses to these viral proteins. Caspase 3/7 activity, a marker of apoptosis, increased in a dose-dependent manner following EP treatment, as assessed by the Caspase-Glo 3/7 assay (Figure 2C). EP significantly inhibited cell proliferation within 2 h of treatment, and cytotoxicity escalated over time (Figure 2D,E).

### 2.3. Effect of VK2 on SARS-CoV-2 Ep-Induced Cytotoxicity

VK2, a menaquinone with known anti-inflammatory properties [18], was evaluated for its ability to modulate EP-induced cytotoxicity in K562 cells. VK2 treatment mitigated EP-induced suppression of cell proliferation (Figure 3A). Furthermore, VK2 reduced caspase 3/7 activity, cytotoxicity, and the induction of apoptosis (Figure 3B–D). VK2 also attenuated EP-induced expression of cytokine genes, including *IL-1α*, *IL-1β*, and *IL-6* (Figure 3E). Specifically, while EP treatment upregulated TNF expression, VK2 effectively suppressed this increase, underscoring the potential anti-inflammatory effects of VK.

### 2.4. Anti-Leukemic Effect of VK2 on CML Cells

The anti-tumor properties of VK2 have been explored in various cancers [18]. In this study, we investigated the anti-leukemic potential of VK2 in CML cell lines K562 and K562 PR. VK2 inhibited the proliferation of both cell lines in a dose-dependent manner (Figure 4A), with dose-dependent cytotoxicity also observed (Figure 4B). These results suggest that VK2 may have therapeutic potential in CML. In a colony formation assay, VK2 significantly reduced the number of colonies formed by K562 cells, indicating its inhibitory effects on cell proliferation and survival (Figure 4C).

### 2.5. Synergistic Effects of Imatinib and VK2 in CML Cells

Finally, we assessed the combined efficacy of imatinib and VK2 in K562 and K562 PR cells. Co-treatment with imatinib and VK2 resulted in a marked reduction in cell proliferation (Figure 5A). Both caspase 3/7 activity and cytotoxicity were significantly increased with co-treatment (Figure 5B,C). Mitochondria, which play a pivotal role in cellular bioenergetics and homeostasis, showed reduced mitochondrial membrane potential (MMP, ΔΨm), a key indicator of mitochondrial dysfunction [20], in cells treated with the combination of imatinib and VK2 (Figure 5D). A colony formation assay also revealed a substantial decrease in both the number and size of colonies following co-treatment (Figure 5E,F). These findings suggest that imatinib and VK2 have synergistic anti-leukemic effects in CML cells.

## 3. Discussion

The SARS-CoV-2 virus precipitated a global pandemic with profound consequences for public health worldwide [4]. Identifying effective therapeutic agents remains critical, particularly those targeting the viral structural proteins essential for viral replication and pathogenesis [21]. Among these, the EP of SARS-CoV-2 is crucial for multiple stages of the viral life cycle and contributes to the virus’s virulence [21]. As a highly conserved structural protein across coronaviruses, the EP represents a promising target for antiviral therapies [21].

Historically, cell death was perceived as an isolated, autonomous process. However, recent advancements in molecular biology have revealed intricate interconnections among various cell death pathways, highlighting the existence of crosstalk and compensatory mechanisms. Apoptosis, a form of programmed cell death, is characterized by hallmark features such as cell shrinkage, nuclear condensation, chromosomal DNA fragmentation, membrane blebbing, and apoptotic body formation [22]. SARS-CoV-2 EP induces apoptosis through diverse signaling pathways, further underscoring its role in viral pathogenesis.

A prior study reported reduced circulating levels of VK1 and VK2 (MK-7) alongside elevated MK-4 levels in patients with COVID-19 compared to healthy controls [23]. The significant depletion of VK2 (MK-7) in patients with COVID-19, regardless of the disease severity, suggests that COVID-19 may drive consumption of this vitamin subtype, likely due to the inflammatory and oxidative stress associated with the disease [23]. Murdaca et al. reported that vitamin D supports immune functions, reducing the severity of COVID-19 infection. Vitamin D modulates innate immunity, enhances adaptive immunity, and potentially mitigates virus-induced cytotoxicity, thus playing a crucial role in both disease prevention and severity reduction [24]. VK2 possesses a longer half-life than VK1 [25], with VK1 primarily localized to hepatic tissues due to its role in coagulation, while VK2 is distributed in the circulation and extrahepatic tissues. VK2 is known to regulate bone metabolism, with potential implications in cancer treatment [15]. Additionally, VK2 has been associated with hepatic dysfunction, neurodegenerative disorders, obesity, and immune modulation [23]. Alterations in VK metabolism may function as a crucial mechanism linking COVID-19-induced lung injury to the development of thromboembolism. VK2’s protective role against SARS-CoV-2-induced cytotoxicity may attenuate vascular and pulmonary complications associated with COVID-19, necessitating further investigation. Furthermore, our results highlight the immunomodulatory role of VK2 in suppressing pro-inflammatory cytokines such as TNF, IL-1α, and IL-1β. This study used CML cells to model viral cytotoxicity and explore the broader viral effects on malignancy. VK2’s enhancement of imatinib activity and reduction of viral envelope-induced cytotoxicity may seem contradictory but reflect the context-dependent actions of VK2. These findings emphasize the need to consider the dosage and timing in clinical applications, including in healthy subjects. Further research is necessary to elucidate the molecular mechanisms underlying these effects of VK2, which could lead to more targeted therapeutic strategies for viral infections and associated complications, particularly in patients with CML.

Previously, we identified asciminib and VK2 as promising therapeutic agents for ABL1 TKI-resistant CML cells, particularly due to their potential to enhance treatment outcomes [26]. In this study, we observed a decrease in MMP and suppression of colony formation, suggesting that the combination of imatinib and VK2 exerts a synergistic effect on cellular bioenergetics and proliferation. This has significant therapeutic implications, especially for targeting ABL1 TKI-resistant CML, where imatinib remains the first-line treatment. Moreover, the differential cytotoxicity observed between the SARS-CoV-2 S and E proteins in K562 cells emphasizes the importance of dissecting specific viral components to understand their distinct roles in modulating cellular processes. Taken together, these findings demonstrate that VK2 enhances the activity of imatinib and has the potential to mitigate the effects of SARS-CoV-2 infection, suggesting that VK2 could be leveraged as an adjunct therapy to improve the efficacy of treatments targeting ABL1 TKI-resistant leukemic cells.

## 4. Materials and Methods

### 4.1. Reagents

All reagents utilized in this study were of analytical grade. Imatinib, the specific inhibitor of BCR::ABL1, was provided by Novartis Pharma AG (Basel, Switzerland). It was dissolved in dimethyl sulfoxide (DMSO) at a stock concentration of 10 mmol/L and stored at −20 °C in aliquots. Recombinant SARS-CoV-2 S1 Subunit Protein (RBD) with a C-terminal His tag, produced from transfected human HEK293 cells, was purchased from RayBiotech, Inc. (Atlanta, GA, USA). The SARS-CoV-2 EP, containing GST and His tags, was obtained from AcroBiosystems (Newark, DE, USA). VK2 (menaquinone) was sourced from Eisai Co., Ltd. (Bunkyo-ku, Tokyo, Japan) and diluted in the culture medium. All other reagents were procured from Merck KGaA (Darmstadt, Germany).

### 4.2. Cell Lines and Culture

The K562 CML cell line was obtained from the American Type Culture Collection (ATCC, Manassas, VA, USA). The K562 PR (ponatinib-resistant) cell line was established as previously described [27]. All cell lines were maintained in Roswell Park Memorial Institute 1640 (RPMI1640) medium supplemented with 10% fetal calf serum (FCS), 1% penicillin-streptomycin, and 1% glutamine at 37 °C in a humidified 5% CO_2_ atmosphere.

### 4.3. Data Collection and Processing

To assess the differentially expressed transcripts, mRNA profiles of peripheral blood mononuclear cells (PBMCs) from five patients with chronic phase CML and five healthy controls were obtained via Illumina NextSeq 500 sequencing (GSE100026). This dataset was sourced from the GEO database (https://www.ncbi.nlm.nih.gov/geo/query/acc.cgi?acc=GSE100026, accessed on 22 May 2024) [28]. Transcriptomic profiles of PBMCs from patients with COVID-19 (mild and severe cases) and healthy controls were compared using data from the NanoString platform (GSE227341) [29]. EdgeR, an R package for differential RNA-Seq counts and microarray data analysis, was utilized to process data according to the manufacturer’s instructions. Transcript abundance was calculated using Salmon with the GRCh39 human transcriptome, and data were further analyzed using the tximport package. RNA-Seq data analysis software for academic research was obtained from BxINFO LLC (Shinagawa-ku, Tokyo, Japan). Data visualization was performed using the RNA-Seq open bioinformatics tool (available at http://ranaseq.eu, accessed on 22 May 2024) [30].

### 4.4. Cell Proliferation Assay

K562 and K562 PR cells were seeded at a density of 8 × 10^3^ cells/well in 96-well plates and treated with specified concentrations of S1 protein, EP, and/or VK2 for 24 h. For co-treatment with imatinib and/or VK2, cells were incubated for 72 h. Cell viability was assessed using the CellTiter-Glo™ Luminescent Cell Viability Assay (Promega, Madison, WI, USA), Cell Counting Kit-8 (Dojindo Laboratories, Mashikimachi, Kumamoto, Japan), or the trypan blue exclusion assay (Bio-Rad, Hercules, CA, USA), following the manufacturer’s instructions. Samples were analyzed using an EnSpire Multimode Plate Reader (PerkinElmer, Waltham, MA, USA) or a TC10 Automated Cell Counter (Bio-Rad). Viability was expressed as a percentage relative to the untreated controls.

### 4.5. Caspase 3/7 Activity

Caspase activity was assessed using the Caspase-Glo^TM^ 3/7 Assay Kit (Promega) according to the manufacturer’s protocol. K562 and K562 PR cells were treated with the indicated concentrations of EP for 2–24 h or with imatinib and/or VK2 for 48 h. Luminescence was measured using an EnSpire Multimode Plate Reader (PerkinElmer).

### 4.6. Cytotoxicity Assay

K562 and K562 PR cells were treated with the S1 or E proteins, with or without VK2, for the specified duration. For imatinib and VK2 co-treatment, cells were incubated for 48 h. Cytotoxicity was determined by measuring LDH release using a Cytotoxicity LDH Assay kit (Dojindo Laboratories, Japan) as per the manufacturer’s protocol. LDH released from non-viable cells was quantified with an EnSpire Multimode Plate Reader (PerkinElmer).

### 4.7. Colon Formation Assay

The colony formation potential of K562 and K562 PR cells was evaluated using MethoCult^®^ Express culture medium (Catalog #04437; STEMCELL Technologies, Vancouver, BC, Canada) in accordance with the manufacturer’s protocol. Treated cells were seeded in 6-well plates at a density of 100 cells/well and exposed to imatinib or VK2 for 7–9 days. Colonies were counted using an EVOS™ FL Digital Inverted Fluorescence Microscope (Thermo Fisher Scientific Inc., Waltham, MA, USA).

### 4.8. Apoptosis Assay

K562 cells treated with the specified concentrations of EP and/or VK2 were evaluated for apoptosis using the Annexin V/Propidium Iodide (PI) binding assay (BD Biosciences, Franklin Lakes, NJ, USA) as per the manufacturer’s instructions. Cell suspensions were incubated with Annexin V and PI for 15 min at room temperature in the dark. Sample fluorescence was quantified using flow cytometry (BD Biosciences) by analyzing a minimum of 10,000 cells per sample.

### 4.9. MMP

MMP was assessed using a mitochondrial staining kit (Merck KGaA) in accordance with the manufacturer’s protocol. Following treatment with imatinib (1 µM) and/or VK2 (10 µM) for 48 h, cells were stained with JC-1 monomers, and fluorescence was measured using an EnSpire Multimode Plate Reader.

### 4.10. Statistical Analyses

Statistical analyses were conducted using GraphPad Prism 10 (GraphPad Software, San Diego, CA, USA). Data are presented as mean ± standard deviation (SD), with all experiments conducted in triplicate. Differences between the groups were assessed using the paired Student’s *t*-test. For comparisons involving a control group, one-way analysis of variance (ANOVA) with Dunnett’s post hoc test, was employed. All experiments were performed with an alpha level of 0.05 and n ≥ 3. Statistical significance was set at * *p* < 0.05, ** *p* < 0.01, *** *p* < 0.001, or **** *p* < 0.0001.

## 5. Conclusions

This study underscores the therapeutic potential of co-administering imatinib and VK2 for the management of CML. This combination not only enhances the therapeutic efficacy of imatinib but also has a protective effect against complications associated with COVID-19 in CML cells. These findings open new avenues for optimizing CML treatment and addressing the challenges posed by viral infections in immunocompromised individuals. Further investigation into the molecular mechanisms underlying this synergistic effect, as well as VK2’s protective role against COVID-19, may lead to more effective and comprehensive treatment strategies for CML and other malignancies.

## Figures and Tables

**Figure 1 ijms-25-11800-f001:**
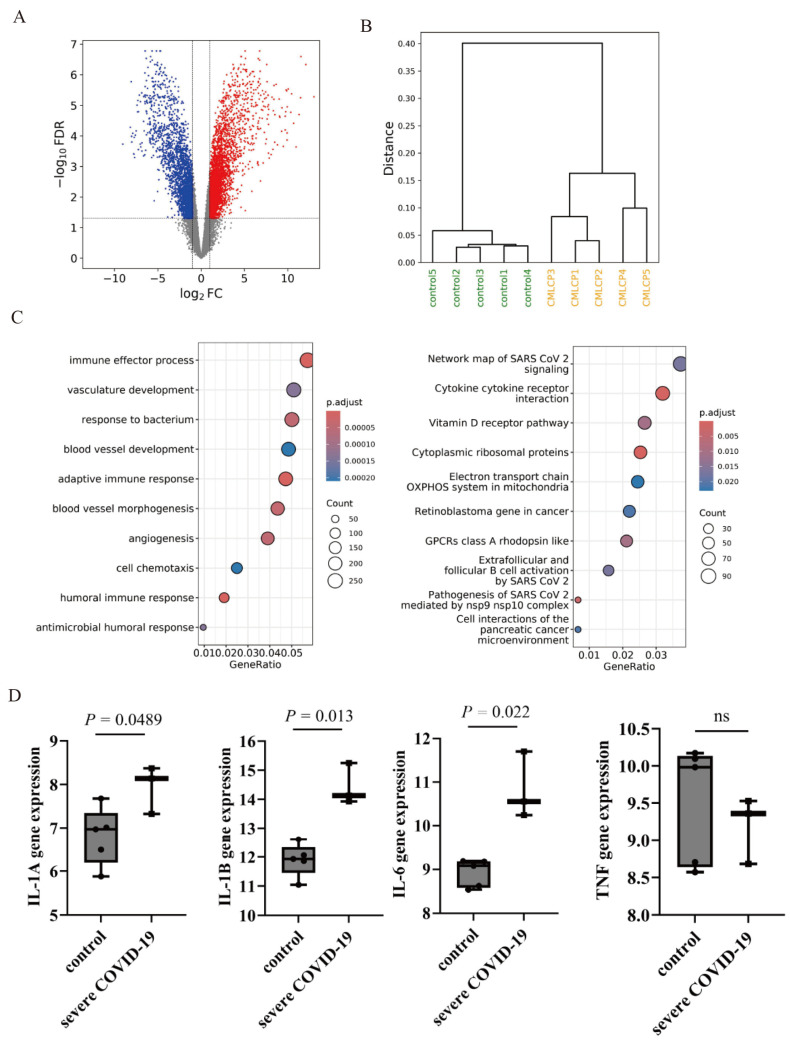
Expression of COVID-19-related genes in chronic myeloid leukemia (CML) cells. (**A**) Volcano plot analysis of datasets retrieved from a public database (GSE100026). (**B**) Dendrogram analysis of datasets retrieved from a public database (GSE100026). (**C**) Gene Ontology (GO) analysis of biological processes and pathway analysis conducted on RNA-Seq data. (**D**) Expression of inflammation-related genes validated using data from the GEO database (GSE227341). ns, not significant.

**Figure 2 ijms-25-11800-f002:**
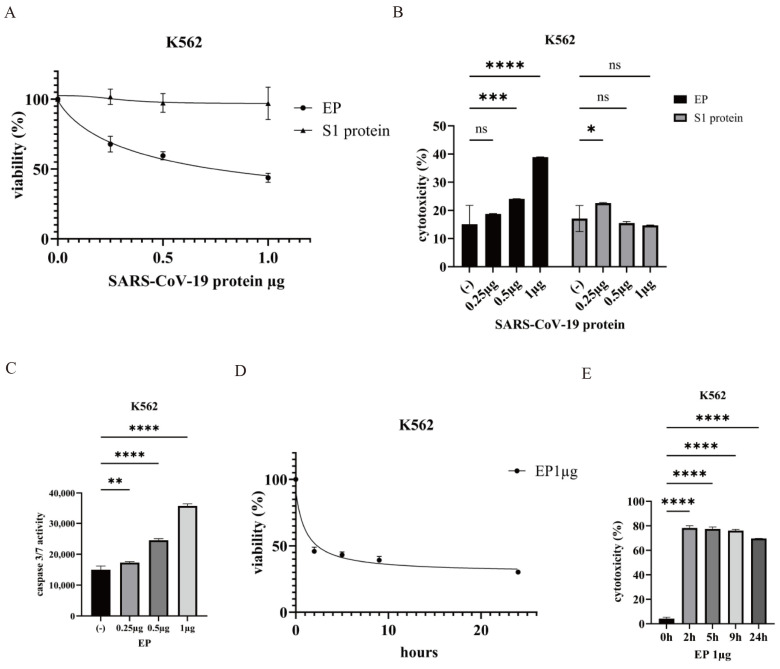
Cytotoxic and apoptotic effects of the SARS-CoV-2 envelope protein on CML cells. (**A**) CML cell lines were cultured with the indicated concentrations of EP or S1 protein for 24 h. Cell growth was evaluated using the Cell Counting Kit-8. (**B**) CML cell lines were cultured in RPMI 1640 medium with 10% fetal calf serum and the indicated concentrations of EP or S1 protein for 24 h. Cytotoxicity was evaluated using the Cytotoxicity LDH Assay Kit. (**C**) CML cell lines were treated with the indicated concentrations of EP for 24 h. Caspase 3/7 activity was measured using the Caspase Glo 3/7 Assay Kit. * *p* < 0.05, ** *p* < 0.01, *** *p* < 0.001, and **** *p* < 0.0001 compared to the control. (**D**,**E**) Time-dependent effects of EP on CML cell lines. Cell growth (**D**) and cytotoxicity (**E**) were evaluated using the Cell Counting Kit-8. ns, not significant.

**Figure 3 ijms-25-11800-f003:**
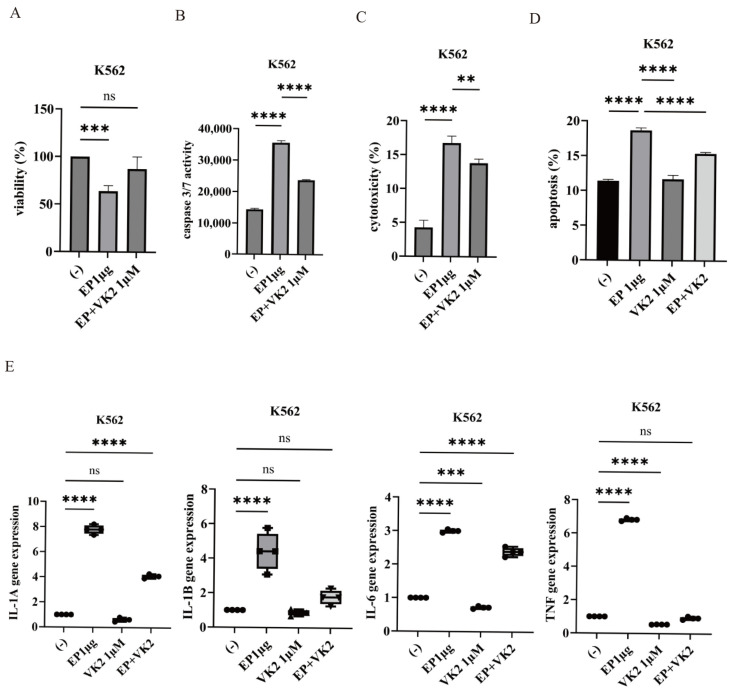
Protective effects of VK2 on CML cells treated with SARS-CoV-2 EP. (**A**–**D**) CML cell lines were cultured with the indicated concentrations of EP and/or VK2 for 24 h. Cell growth (**A**), caspase 3/7 activity (**B**), cytotoxicity (**C**), and apoptosis (**D**) were evaluated. ** *p* < 0.01, *** *p* < 0.001, **** *p* < 0.0001 compared to EP-treated cells. ns, not significant. (**E**) Expression levels of inflammatory-related genes in cells treated with EP and/or VK2 for 24 h. Gene expression was validated using RT-PCR. *** *p* < 0.001, **** *p* < 0.0001 compared to the control. ns, not significant.

**Figure 4 ijms-25-11800-f004:**
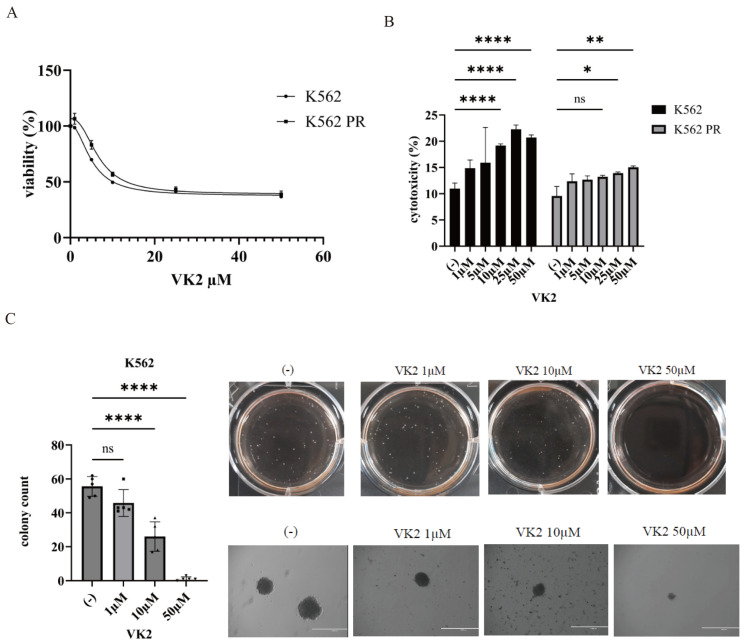
VK2 suppresses CML cell proliferation and colony formation. (**A**,**B**) CML cell lines were incubated with VK2 for 72 h. Cell growth (**A**) and cytotoxicity (**B**) were evaluated. * *p* < 0.05, ** *p* < 0.01, and **** *p* < 0.0001 compared to EP-treated cells. ns, not significant. (**C**) CML cells were treated with the indicated concentrations of VK2 for 7 days. Colonies were photographed with a digital camera and counted using an EVOS™ FL Digital Inverted Fluorescence Microscope (Thermo Fisher Scientific Inc., Waltham, MA, USA). The quantitative graph shows the number of colonies, and representative images are displayed. Scale bar: 1000 μm. **** *p* < 0.0001 compared to the control. ns, not significant.

**Figure 5 ijms-25-11800-f005:**
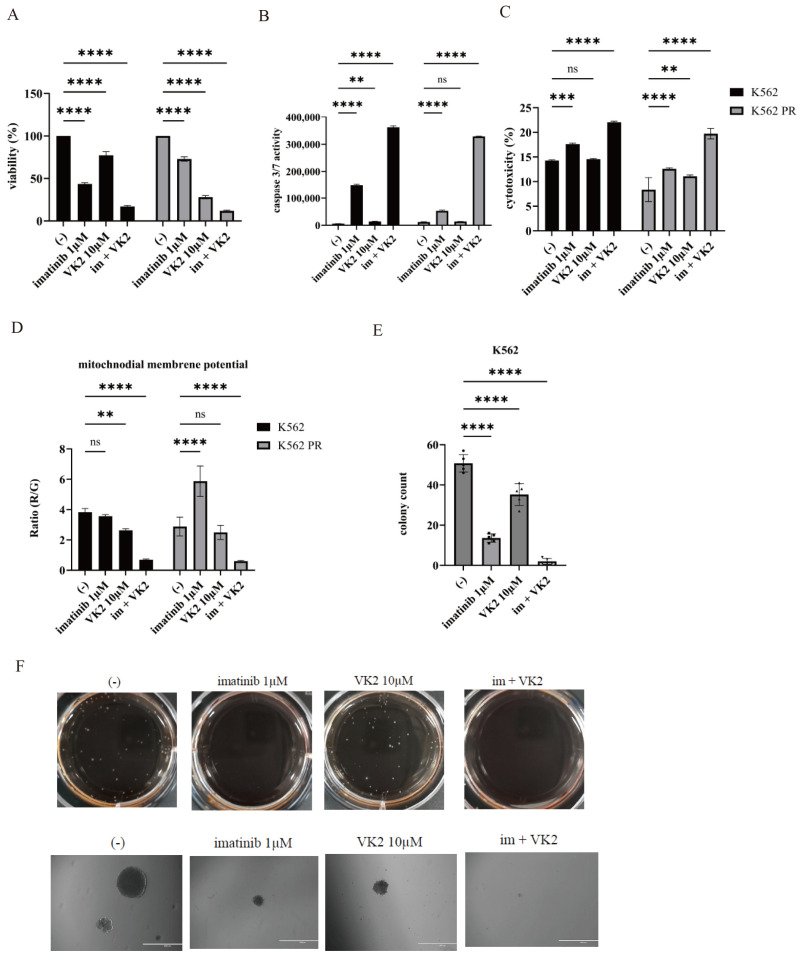
Synergistic effects of imatinib and VK2 in CML cells. (**A**–**C**) CML cell lines were cultured with 1 µM imatinib and/or 10 µM VK2 for 48 h or 72 h. Cell growth (**A**), caspase 3/7 activity (**B**), and cytotoxicity (**C**) were measured. ** *p* < 0.01, *** *p* < 0.001, **** *p* < 0.0001 compared to the control. ns, not significant. (**D**) Mitochondrial membrane potential (MMP) was analyzed in CML cell lines treated with 10 nM asciminib and/or 10 µM VK2 for 48 h using a Mitochondria Staining Kit. ** *p* < 0.01, **** *p* < 0.0001 compared to the control. ns, not significant. (**E**,**F**) Colony formation in CML cells that were treated with VK2 for 7 to 9 days. Colonies were counted (**E**,**F**) photographed using a digital camera and an EVOS™ FL Digital Inverted Fluorescence Microscope. Scale bar: 1000 μm. **** *p* < 0.0001 compared to the control.

## Data Availability

The datasets and materials used and/or analyzed in this study are available from the corresponding author upon reasonable request.

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
