# Peer review of "Vitamin K2 Protects Against SARS-CoV-2 Envelope Protein-Induced Cytotoxicity in Chronic Myeloid Leukemia Cells and Enhances Imatinib Activity"

_ijms, 2024, doi:10.3390/ijms252111800_

Round 1

Reviewer 1 Report

Comments and Suggestions for Authors

In this study, Okabe et al present their investigation of Vit K2 regarding its protective role in SARS-CoV-2 envelope-induced cytotoxicity in CML cells, as well as the enhancement of imatinib activity, as a treatment of CML. This is well-written and concerns an interesting and significant topic. The methodology is adequate and appropriately discussed. In general, I have not detected major issues with the study, but there are some points that need clarification regarding the premisses of the investigators' hypothesis. 

1. The authors suggest that VK2 protects CML cells from SARS-CoV-2 cytotoxicity via the envelope protein. How does this relate with SARS-CoV-2 pathogenesis in CML patients? We would someone want to protect cancer cells from cytoxocity. If CML cells have been used as a cellular model for SARS-Cov-2 cytoxicity in general, how do these evidence apply in healthy individuals, and why the authors specifically allude to the impact of the infection on CML patients? 

2. In relation to my first comment, the authors show that VK2 enhances imatinib activity against CML cell proliferation, but how does this relates to SARS-CoV-2 pathogenesis. Aren't those two separate topics? Especially considering that VK2 reduces envelope-induced cytotoxicity, but on the other hand halts cell proliferation. This effects appear contradictory and the relation between these findings warrants further clarification. 

3. Figures: please make sure that final resolution of the figures is such that allows readability. Font size in the current format is rather difficult to read. 

4. Discussion (lines 198-202): It is not clear how the findings in this study point towards a link to COVID-19 lung injury and thromboembolism. Please elaborate on this statement. 

Comments on the Quality of English Language

I have not detected issues with the quality of English.

Author Response

Response to Reviewer #1

Reviewer Comment:

“In this study, Okabe et al present their investigation of Vit K2 regarding its protective role in SARS-CoV-2 envelope-induced cytotoxicity in CML cells, as well as the enhancement of imatinib activity, as a treatment of CML. This is well-written and concerns an interesting and significant topic. The methodology is adequate and appropriately discussed. In general, I have not detected major issues with the study, but there are some points that need clarification regarding the premisses of the investigators' hypothesis.”

Thank you for your constructive comments.

Reviewer Specific Comment #1: 

“The authors suggest that VK2 protects CML cells from SARS-CoV-2 cytotoxicity via the envelope protein. How does this relate with SARS-CoV-2 pathogenesis in CML patients? We would someone want to protect cancer cells from cytoxocity. If CML cells have been used as a cellular model for SARS-Cov-2 cytoxicity in general, how do these evidence apply in healthy individuals, and why the authors specifically allude to the impact of the infection on CML patients?”

Response to Specific Comment #1:

We agree and describe the discussion in detail.

“This study used CML cells to model viral cytotoxicity and explore the broader viral effects on malignancy. VK2’s enhancement of imatinib activity and reduction of viral envelope-induced cytotoxicity may seem contradictory but reflect the con-text-dependent actions of VK2. These findings emphasize the need to consider the dosage and timing in clinical applications, including in healthy subjects (page 8 line 218-223).”

Reviewer Specific Comment #2: 

“In relation to my first comment, the authors show that VK2 enhances imatinib activity against CML cell proliferation, but how does this relates to SARS-CoV-2 pathogenesis. Aren't those two separate topics? Especially considering that VK2 reduces envelope-induced cytotoxicity, but on the other hand halts cell proliferation. This effects appear contradictory and the relation between these findings warrants further clarification.”

Response to Specific Comment #2:

We agree and describe in the text.

“This study used CML cells to model viral cytotoxicity and explore the broader viral effects on malignancy. VK2’s enhancement of imatinib activity and reduction of viral envelope-induced cytotoxicity may seem contradictory but reflect the con-text-dependent actions of VK2. These findings emphasize the need to consider the dosage and timing in clinical applications, including in healthy subjects. Further research is necessary to elucidate the molecular mechanisms underlying these effects of VK2, which could lead to more targeted therapeutic strategies for viral infections and associated complications, particularly in patients with CML. (page 8 line 218-225).”

Reviewer Specific Comment #3: 

“Figures: please make sure that final resolution of the figures is such that allows readability. Font size in the current format is rather difficult to read.”

Response to Specific Comment #3:

We agree and the font size for the Figure has been enlarged to make it easier to read.

Reviewer Specific Comment #4: 

“Discussion (lines 198-202): It is not clear how the findings in this study point towards a link to COVID-19 lung injury and thromboembolism. Please elaborate on this statement.”

Response to Specific Comment #4:

We agree and describe in the text.

“Alterations in VK metabolism may function as a crucial mechanism linking COVID-19-induced lung injury to the development of thromboembolism. VK2's protective role against SARS-CoV-2-induced cytotoxicity may attenuate vascular and pulmonary complications associated with COVID-19, necessitating further investigation (page 8 line 213-217).”

Reviewer 2 Report

Comments and Suggestions for Authors

The paper is interesting and well written. The authors investigated the effects of SARS-CoV-2 proteins on chronic myeloid leukemia cell (CML) viability and the protective role of vitamin K2 (VK2) in conjunction with imatinib. The study confirmed that VK2 protects CML cells from SARS-CoV-2-induced cytotoxicity and enhances the therapeutic efficacy of imatinib. I suggest to disccus the role of vitamin D and microbioma on immune responses (see and add as referecnces papers by Murdaca et al concerning vitamin D, microbioma and immune responses).

Comments on the Quality of English Language

Minro english editing

Author Response

Response to Reviewer #2

Reviewer Comment:

“The paper is interesting and well written. The authors investigated the effects of SARS-CoV-2 proteins on chronic myeloid leukemia cell (CML) viability and the protective role of vitamin K2 (VK2) in conjunction with imatinib. The study confirmed that VK2 protects CML cells from SARS-CoV-2-induced cytotoxicity and enhances the therapeutic efficacy of imatinib.”

Thank you for your constructive comments.

Reviewer Specific Comment #1: 

“I suggest to discuss the role of vitamin D and microbioma on immune responses (see and add as references papers by Murdaca et al concerning vitamin D, microbioma and immune responses).”

Response to Specific Comment #1:

We agree and add the reference and describe the text in detail.

“Murdaca et al. reported that vitamin D supports immune functions, reducing the severity of COVID-19 infection. Vitamin D modulates innate immunity, enhances adaptive immunity, and potentially mitigates virus-induced cytotoxicity, thus playing a crucial role in both disease prevention and severity reduction [24] (page 8 line 204-208).”